# Meta-iAVP: A Sequence-Based Meta-Predictor for Improving the Prediction of Antiviral Peptides Using Effective Feature Representation

**DOI:** 10.3390/ijms20225743

**Published:** 2019-11-15

**Authors:** Nalini Schaduangrat, Chanin Nantasenamat, Virapong Prachayasittikul, Watshara Shoombuatong

**Affiliations:** 1Center of Data Mining and Biomedical Informatics, Faculty of Medical Technology, Mahidol University, Bangkok 10700, Thailand; nalini.sch@mahidol.edu (N.S.); chanin.nan@mahidol.edu (C.N.); 2Department of Clinical Microbiology and Applied Technology, Faculty of Medical Technology, Mahidol University, Bangkok 10700, Thailand; virapong.pra@mahidol.ac.th

**Keywords:** therapeutic peptides, antiviral peptide, classification, machine learning, random forest, meta-predictor

## Abstract

In spite of the large-scale production and widespread distribution of vaccines and antiviral drugs, viruses remain a prominent human disease. Recently, the discovery of antiviral peptides (AVPs) has become an influential antiviral agent due to their extraordinary advantages. With the avalanche of newly-found peptide sequences in the post-genomic era, there is a great demand to develop a sequence-based predictor for timely identifying AVPs as this information is very useful for both basic research and drug development. In this study, we propose a novel sequence-based meta-predictor with an effective feature representation, called Meta-iAVP, for the accurate prediction of AVPs from given peptide sequences. Herein, the effective feature representation was extracted from a set of prediction scores derived from various machine learning algorithms and types of features. To the best of our knowledge, the model proposed herein represents the first meta-based approach for the prediction of AVPs. An overall accuracy and Matthews correlation coefficient of 95.20% and 0.90, respectively, was achieved from the independent test set on an objective benchmark dataset. Comparative analysis suggested that Meta-iAVP was superior to that of existing methods and therefore represents a useful tool for AVP prediction. Finally, in an effort to facilitate high-throughput prediction of AVPs, the model was deployed as the Meta-iAVP web server and is made freely available online at http://codes.bio/meta-iavp/ where users can submit query peptide sequences for determining the likelihood of whether or not these peptides are AVPs.

## 1. Introduction

Human morbidity, mortality, and economic productivity continue to be affected by viral infections and their associated diseases. The dominance of sporadic viral outbreaks by zoonotic viruses such as Ebola and Zika in recent years have added to the prevalence of viral species with which humans are already in battle (i.e., human immunodeficiency virus (HIV), rhinoviruses, and influenza viruses). Viruses are successful in causing malaise to humans due to their high genetic variation, different routes of transmission, efficient replication, and the capability to persist in the host cells [1]. Furthermore, according to the global threat list of 2019 as compiled by the WHO, virus infections were seen to dominate [2]. Although, up until recently, trial and error has led to the discovery of 90 antiviral drugs approved for the treatment of 9 virus families (i.e., HIV, hepatitis B virus, hepatitis C virus, human cytomegalovirus, influenza virus, herpes virus, varicella-zoster virus, respiratory syncytial virus, and human papillomavirus), these drugs cannot begin to cover the >200 viruses discovered thus far [3,4]. In addition, major breakthroughs in combating viral infections by vaccine production have led to remarkable advances in modern medicine such as the eradication and control of disease such as small pox [5] and polio [6], respectively. Nevertheless, the development of new vaccines remains a huge challenge in terms of time and expenses [7]. Unfortunately, the ever-increasing reports of antiviral resistance [8,9,10] coupled with the emergence and re-emergence of viral epidemics as observed for H1N1 [11], Ebola [12], and Zika [13] viruses, demands the production of new antiviral drugs with broad-spectrum activity [14]. More recently, peptide-based drugs have gained much interest as a new class of drugs due to their ability to be highly selective, relatively safe while also possessing good tolerability and a lower production cost [15]. Besides the advantages of peptide-based drugs, a short half-life, immunogenic potential, and low oral absorption are some of their current limitations [16].

Antiviral peptides (AVPs) is a subset belonging to the group of antimicrobial peptides (AMPs) and in that regard, exhibits antiviral activity. As of 23 September, 2019, the antimicrobial database (APD3) contains a total of 3129 AMPs, out of which, 188 are antiviral peptides [17]. Similarly, another database of antimicrobial peptides, DRAMP 2.0, contains 19,899 entries which consist of general, patent, and clinical AMPs [18,19]. In addition, a database focused solely on antiviral peptides contains 2683 experimentally verified AVPs including 624 modified AVPs [1]. Additionally, there are other databases that focus on the structure and antimicrobial activity of natural and synthetic peptides [20] as well as therapeutic peptides [21,22,23]. Thus, it is evident that peptide-based research is gaining momentum. In some cases, a given peptide shows more than one activity and is, therefore, called a promiscuous peptide (i.e., showing dual antimicrobial and antiviral effects). In addition, AVPs have been shown to possess cationic and amphipathic characteristics with positive net charges, all of which are essential for these peptides to work as antimicrobials [24]. Moreover, hydrophobicity seems to be a key property for peptides with activity against enveloped viruses [25,26]. To date, Fuzeon™ (Enfuvirtide), a synthetic peptide that blocks viral fusion by binding to gp41 (polypeptide chain) of HIV type-1 envelope protein is the only peptide to have been commercialized [27]. In addition, Bulevirtide™ (Myrcludex B), an anti-Hepatitis B and Hepatitis D peptide targeting sodium taurocholate co-transporting polypeptide (NTCP) of liver cells, has also been studied in a phase IIb clinical trial [28] and is scheduled for phase III trials [29]. The structure of some AVPs that have already been elucidated experimentally, are shown in Figure 1.

Furthermore, there are several mechanisms of action whereby antiviral therapeutic agents can inhibit viral activity (i.e., block the attachment of viruses, prevent fusion of viruses to host cells, interrupt the signaling process of viruses, or inhibit the replication of viruses in host cells) [30]. Currently, some studies have shown that AVPs inhibit the fusion of viruses to host cells [14,31,32]; while others have shown that AVPs interfere with viral replication [33,34,35] and attachment of the virus to host cells [36,37,38]. For example, P9, an AVP derived from mouse β-defensin acts against various flu strains (i.e., H1N1, H3N2, H5N1, H7N7, and H7N9) by binding to viral glycoproteins and inhibiting RNA replication through the prevention of viral fusion in the endosome [39]. Additionally, protegrin-1, a cyclical cationic peptide derived from swine white blood cells, showed potent antiviral activity against dengue virus by inhibiting the specific viral protease important for dengue virus replication, named NS2B-NS3pro [40]. Hence, accurately identifying the biological activities of peptides provides great importance for the exploration of the mechanism of action of AVPs and the development of antiviral drugs. However, the experimental approaches are still very slow, inefficient, and expensive. Besides, with the rapid explosion of newly-found peptide sequences in the post-genomic era, the peptide sequences in various database are rapidly increasing day by day. In that regard, bioinformatics-based tools are crucial for efficient analysis of the ever-increasing availability of data. Thus, it is in a great demand to develop a prediction model based on an efficient machine learning algorithm for fast and reliably identifying the biological activities of peptides according to their primary sequences. This process could further shed light on novel AVPs having potent clinical outcomes.

Until now, there are four prediction models based on various machine learning (ML) algorithms that have been developed for AVP prediction, i.e., AVPpred [41], Chang et al.’s method [42], Zare et al.’s method [43], and AntiVPP 1.0 [44]. Three of the four prediction models [41,42,44] were performed on the same benchmark datasets, as summarized in Table 1. Initially, Thakur et al. [41] was the first to propose a prediction model for AVP prediction called AVPred as well as established the two benchmark datasets T^544p+407n^ + V^60p+45n^ and T^544p+544n^ + V^60p+60n^. AVPred was constructed by using a support vector machine (SVM)-based model with physicochemical properties from the AAindex database. AVPred provided moderate prediction accuracies on the independent datasets V^60p+45n^ and V^60p+60n^ of 85.7% and 92.5%, respectively. Shortly afterward, Chang et al. [42] utilized a combination feature of amino acid composition (AAC) and aggregation tendencies to develop a random forest (RF) model. Their prediction model achieved higher prediction accuracies as compared to AVPred with 89.5% and 93.3% for T^544p+407n^ + V^60p+45n^ and T^544p+544n^ + V^60p+60n^ datasets, respectively. Recently, Lissabet et al. [44] proposed a computation tool based on RF in conjunction with various physicochemical properties called AntiVPP 1.0. In their experimental setting, AntiVPP 1.0 was developed using one of the two benchmarked datasets, i.e., T^544p+544n^ + V^60p+60n^ and obtained a prediction accuracy of 93.0% which did not show any improvement as compared to AVPpred and Chang et al.’s method. Although, the above-mentioned methods produced promising results, there is still room for improvement in regards to prediction performance. First, the features used for constructing the previous methods did not offer the sequence-order or position-specific information and hence might considerably limit the prediction quality. Second, most of the existing predictors [41,42,44] were developed using the embodiment of redundant features, causing a decrease in performance. Finally, the accuracy and transferability of the prediction model still require improvement.

Motivated by the aforementioned issues, we proposed a novel sequence-based meta-predictor, called Meta-iAVP, for the prediction of AVPs from given peptide sequences to address the shortcomings of the existing methods. First, the benchmark datasets were collected to construct a model and fairly compare with the previous models. Second, we encoded the peptide sequence with AAC, pseudo amino acid composition (PseAAC), amphiphilic pseudo amino acid composition (Am-PseAAC), dipeptide composition (DPC), and g-gap dipeptide composition (GDC). Third, we fed each feature separately into six different ML algorithms, i.e., RF, SVM, *k*-nearest neighbor (*k*-NN), recursive partitioning and regression trees (rpart), generalized linear model (glm), and extreme gradient boosting (XGBoost), to generate a new feature representation. Subsequently, effective feature representation was used to build a meta-predictor. The performance comparisons on the two benchmark datasets illustrated that Meta-iAVP significantly outperformed other existing AVP predictors. To the best of our knowledge, our proposed model is the first meta-based approach in the prediction of AVPs. We anticipate that Meta-iAVP may serve as a useful computational resource for high-throughput AVP prediction and also facilitate experimental researchers in the discovery of novel AVPs. Finally, for the convenience of experimental scientists, a Meta-iAVP web server was established and made freely available online at http://codes.bio/meta-iavp/.

## 2. Results

In this study, AVPs and Non-AVPs were predicted by the proposed method Meta-iAVP. Firstly, the importance of each amino acid to antiviral activities of peptides using mean decrease of Gini index (MDGI) and univariate analysis were performed. Secondly, the features that are beneficial for discriminating AVPs from Non-AVPs were determined by conducting performance comparisons between five types of features, i.e., AAC (20D), DPC (400D), GDC (400D), PseAAC (20 + 2λ), and Am- PseAAC (20 + 2λ), and six commonly used ML algorithms. Thirdly, Meta-iAVP based on the meta-predictor was constructed by using the new feature representation as the input feature. Finally, to serve easy and rapid classification of query peptide sequence, Meta-iAVP is exploited as a free prediction web server for discriminating AVPs and Non-AVPs. Figure 2 summarizes the workflow of the computational approach of Meta-iAVP.

### 2.1. Biological Space of Antiviral Peptides

As previously mentioned, AAC and DPC descriptors allow us to decipher the biochemical and biophysical properties of antiviral peptides. Preceding studies have used the AAC and DPC as to gain further insights on the characterization of therapeutic peptides [45,46,47,48] and various protein functions [49,50,51,52]. In this study, the value of MDGI was adopted to rank and estimate the importance of each AAC and DPC feature. Table 2 and Table 3 list the percentage values of the top 20 amino acids for both AVPs and Non-AVPs as derived from experimental validation and random datasets, respectively. In addition, a heatmap showing the feature importance for DPC is provided in Figure 3. From Table 2 and Table 3, it can be observed that the ten informative amino acids with the highest MDGI values are Lys, Thr, Leu, Ile, Ser, Trp, Asn, Arg, Cys, and Glu (49.27, 46.27, 35.06, 34.52, 30.95, 30.93, 30.19, 28.52, 26.33, and 24.87, respectively) and Lys, Pro, Cys, Thr, Ser, Trp, Val, Ala, Gly, and Leu (77.11, 68.87, 57.68, 46.84, 39.57, 36.83, 25.69, 24.40, 24.25, and 23.80, respectively) for the experimental validation and random Non-AVP datasets, respectively. Meanwhile, Figure 3a,b shows that the five top-ranked dipeptides according to their MDGI value are LL, RK, LV, WI, and EI for the experimentally validated dataset (T^544p+407n^dataset) and KR, KK, GP, AS, and SA for the random Non-AVP dataset (T^544p+544n^ dataset), respectively.

Interestingly, three of the five top-ranked informative amino acids from both Table 2 and Table 3, are common and represent polar amino acids (i.e., Lys, Thr, and Ser), while the other amino acids are non-polar and hydrophobic residues (i.e., Leu and Ile for the experimental dataset and Pro for the random Non-AVP dataset). As stated, the top ranked amino acid, Lysine (Lys) was observed in both the experimentally validated dataset as well as the random Non-AVP dataset. Being a basic residue, Lys is abundantly found in the composition of therapeutic peptides due to its ability to enhance the electrostatic properties that facilitate the interaction and insertion of peptides into the anionic cell walls and phospholipid membranes of microorganisms [53]. Thus, the cationic role of Lys is observed in various AMPs which also function as AVPs. For instance, first published in 1986, the study by Daher et al. [54] reported the antiviral role of a cationic peptide, α-defensin which was described as inhibiting a number of viruses including herpes simplex virus types one and two, cytomegalovirus as well as inhibiting the vesicular stomatitis virus with human neutrophil peptide 1 (HNP1) in vitro. Since then, many reports have shown antiviral activity of cationic host-defense peptides such as α-defensins (i.e., HNP-1, HNP-2, HNP-3, and HNP-4), β-defensins (i.e., HBD-2 and HBD-3), and θ-defensin (i.e., Retrocyclin-2), and the use of effective antiviral therapy with cathelicidins (i.e., LL-37), as previously reviewed [36,55,56,57,58,59]. Furthermore, Mandelboim et al. observed that the initiation of lysis via natural killer cells by the P8 epitope of coxsackie viral peptide was pronounced with Lys as compared to other basic amino acid residues such as Arg or His [60]. Hence, the role of Lys in providing cationic properties to a given peptide sequence is fundamental and leads to the enhancement of its antiviral activities.

Threonine (Thr) is another common amino acid observed between the two datasets of Table 2 and Table 3. Thr plays an essential role in the phosphorylation of virus-encoded serine/threonine kinases, a unique feature of large DNA viruses [61]. This important phosphorylation usually results in a functional change of the target protein by interfering with its enzymatic activity, cellular location, and/or association with other proteins [61]. Therefore, a disruption of this property could hinder the efficient spread of the virus. This notion was also elucidated in a study conducted by Santos et al. [62] on a nuclear shuttle protein (NSP)-interacting kinase (NIK1) which acts as a receptor-like kinase identified as a virulence target of the begomovirus NSP. The authors conducted mutagenesis on residues Thr-474 and Thr-468 on the A-loop of the NIK1 and observed that these mutations impaired autophosphorylation and were unable to attain kinase activation. In addition, Hale et al. [63] reported that an Ala substitution of Thr-215 of the NS1 protein phosphorylation mechanism caused a disruption in viral propagation of human influenza A virus. Similarly, Hemonnot et al. [64] conducted mutational analysis of HIV mitogen-activated protein (MAP) kinase extracellular signal-regulated kinase-2 (ERK-2) by substitution of Thr-23 to Ala-23. The resulting electron microscopy and western blot analysis showed that the substitution of a single Thr-23 residue, which provided an essential function in the release of viral particles from the cell surface, was disrupted. Thus, from the aforementioned studies, it is clear that Thr is extremely vital for proper kinase phosphorylation of viral proteins which further allow for efficient viral budding from infected cells.

The third most important amino acid observed from Table 2 and Table 3 was Serine (Ser) which plays an essential role in several cellular and metabolic processes [65]. In addition, as previously mentioned, Ser also makes up an important component of virus-encoded serine/threonine kinases [61]. Furthermore, an extensively studied and well-known AMP, lactoferrin, is recognized as a potent inhibitor of various viruses such as human immunodeficiency virus, herpes simplex virus types one and two, human cytomegalovirus, hepatitis C virus, hepatitis B virus, and respiratory syncytial virus. [66]. One such study conducted by Scala et al. [66], examined in detail the structure of lactoferrin-derived peptides and their activity against influenza virus using protein-protein interactions. In addition, all the peptide fragments tested were derived from the Ser418-Pro429 loop which formed a structural conformation that was critical for the resulting peptide activity. The authors noted that the presence of Ser was observed in the top three active peptide fragments. Hence, the presence of Ser in terms of formation of effective peptides for antiviral activity is highly advantageous.

### 2.2. Performance Comparison of Various Types of Features

To assess the effectiveness of each feature in discriminating AVPs from Non-AVPs, the five-fold CV and independent validation test were conducted for each feature by performing six commonly used ML models. Figure 4 and Figure 5 provide the performance comparisons over the five-repeated five-fold CV and independent test results on T^544p+407n^ + V^60p+45n^ and T^544p+544n^ + V^60p+60n^ datasets, respectively. As seen in Figure 4 and Figure 5, the average Ac over the five-repeated five-fold CV on T^544p+407n^ and T^544p+544n^ datasets are (78.52%, 78.72%, 79.69%, 78.68%, and 77.04%) and (84.91%, 84.88%, 85.28%, 82.19%, and 86.44%) for ACC, PseAAC, Am-PseAAC, DPC, and GDC, respectively. The average Ac of each type of feature was obtained by averaging six Ac values derived from six ML algorithms over the five-repeated five-fold CV and independent validation test. Meanwhile, the performance comparisons on the independent validation datasets V^60p+45n^ and V^60p+60n^ were (80.29%, 83.17%, 79.01%, 79.49%, and 77.41%) and (86.16%, 86.44%, 85.88%, 86.02%, and 84.59%) for ACC, PseAAC, Am-PseAAC, DPC, and GDC, respectively. For performance comparisons among the six ML models, the prediction results showed that average Ac over the five-repeated five-fold CV and independent test results on T^544p+407n^ + V^60p+45n^ and T^544p+544n^ + V^60p+60n^ datasets were (80.55%, 76.36%, 74.46, 86.86%, 85.93%, and 85.65%) and (82.16%, 78.00%, 74.09%, 86.86%, 85.93%, and 85.65%), respectively.

By observing the performance comparisons in Figure 4 and Figure 5, it could be summarized as follows: (i) ACC and DPC features did not afford better performance than other three predictors but they provide more interpretability for discriminating AVPs from Non-AVPs, which is helpful for biologists in designing novel peptides. This observation is quite consistent with previous works [41,42]; (ii) the top three most powerful ML models over the five-repeated five-fold CV and independent test are RF, XGBoost, and SVM; and (iii) these prediction results demonstrate that the three top-ranked important features in discriminating AVPs from Non-AVPs are PseAAC, AAC, and DPC, where AAC and PseAAC are the most beneficial features for discriminating AVPs from Non-AVPs on the benchmark datasets T^544p+407n^ + V^60p+45n^ and T^544p+544n^ + V^60p+60n^, respectively.

### 2.3. Construction of the Meta-iAVP Model

In general, the meta-predictor utilizes an important pattern from the predicted output derived from different predictors under the assumption that using combined methods will provide substantially accurate prediction results than a single method [67,68,69,70,71]. As described above, AAC and PseAAC are the most important features for discriminating AVPs from Non-AVPs. Thus, to verify the power of these two features in AVP prediction, the six ML models are trained with the AAC and PseAAC features for performing on the benchmark datasets T^544p+407n^ + V^60p+45n^ and T^544p+544n^ + V^60p+60n^, respectively, and their performance comparisons are listed in Table 4. Amongst the six ML models, Table 4 shows that the RF model with the AAC feature performs best with the highest Ac, Sn, Sp, and MCC of 86.54%, 86.54%, 86.36%, and 0.73, respectively, over the independent validation test on V^60p+45n^ dataset. Meanwhile, the RF model with the PseAAC feature shows superiority in discriminating AVPs from Non-AVPs on the dataset V^60p+60n^ with the highest Ac, Sn, Sp, and MCC of 91.53%, 90.00%, 93.10%, and 0.83, respectively. Therefore, the AAC and PseAAC features were used as the initial features for constructing the new feature representation to train the meta-predictor, as summarized in the Section 3.6.

To demonstrate the superiority and capability of our proposed model, we compared the aforementioned prediction results with the meta-predictor. Table 4 shows that the overall Ac and MCC values obtained from the meta-predictor are 4–9% and 9–17%, respectively, which are higher than those resulting from *k*-NN, rpart, glm, RF, XGBoost, and SVM models on both V^60p+45n^ and V^60p+60n^ datasets. It could be stated that our proposed meta-predictors are justified as the more powerful and highly efficient AVP predictor. For convenience of the subsequent description, we will refer to these two meta-predictors as Meta-iAVP.

### 2.4. Analysis of new feature representation

As seen in Figure 4, Figure 5 and Table 4, the improved performances of the proposed model was achieved due to the method that takes new feature representation as the input feature and the meta-predictor as the prediction engine. In the previous sub-section, the AAC and PseAAC were mentioned as the optimum features amongst the five popular-used features, thus, these two features were used to compare with the new feature representation. To demonstrate the effectiveness of the new feature representation, the principle component analysis (PCA) approach is used to compare the distribution of AVPs (red circles) and Non-AVPs (blue circles) by representing them with PCA scores as illustrated in Figure 6. In this study, PCA analysis was performed using the FactorMineR R package [72] in R programing environment. To perform PCA analysis, T^544p+407n^ + V^60p+45n^ and T^544p+544n^ + V^60p+60n^ datasets were represented by the first two PCs (PC1 and PC2), where the percentage of variance can be explained by the first two PCs where high percentage values is suggestive of the feature importance for the predictive model. Figure 6a,c depict the distribution of AAC and a new feature representation, respectively, obtained from the dataset T^544p+407n^ + V^60p+45n^, while Figure 6b,d represent the distribution of PseAAC and a new feature representation, respectively, obtained from T^544p+544n^ + V^60p+60n^ dataset. It should be noted that, more overlap between the red and blue circles indicate the feature is less capable in AVP prediction. Remarkably, Figure 6c,d revealed that the new feature representation is efficient and effective as the input feature for discriminating AVPs from Non-AVPs. This might explain why the proposed model, Meta-iAVP, outperformed the other conventional models.

### 2.5. Comparison of Meta-iAVP with the State-of-Art Predictors

To indicate the effectiveness of Meta-iAVP, we benchmarked it against the three state-of-art AVP predictors namely AVPpred [41], Chang et al.’s method [42], and AntiVPP 1.0 [44]. Among the three AVP predictors, only AVPpred and Chang et al.’s method provided the prediction results over five-fold CV and independent test results on T^544p+407n^ + V^60p+45n^ and T^544p+544n^ + V^60p+60n^. In view of this, we only performed comparisons between Meta-iAVP with AVPpred and Chang et al.’s method. The overall performance comparisons of Meta-iAVP with other three existing methods over five-fold CV and independent test results on T^544p+407n^ + V^60p+45n^ and T^544p+544n^ + V^60p+60n^ are shown in Table 5. The pioneer work on the benchmark datasets was firstly reported by Thakur et al. [41]. Initially, they provided prediction results (Ac, MCC) on the independent dataset V^60p+45n^ and V^60p+60n^ with (85.70%, 0.71) and (92.50%, 0.85), respectively. Later on, Chang et al. [42] utilized the RF model cooperating with their proposed features to enhance the prediction performance. Their prediction model yielded (89.50%, 0.79) and (93.30%, 0.87) on the independent datasets V^60p+45n^ and V^60p+60n^, respectively, indicating that Chang et al.’s method outperformed AVPpred. Meanwhile, as noticed in Table 5, our proposed model Meta-iAVP achieved the best performances in terms of Ac, Sn, and MCC (V^60p+45n^, V^60p+60n^) of (95.20%, 94.90%), (93.20%, 98.30%), and (0.90, 0.90), respectively. Remarkably, Ac and MCC of Meta-iAVP were approximately 3.3–11.0% and 3.0–11.0% higher than the three state-of-art AVP predictors, thus demonstrating the superiority of our proposed predictor.

With regard to the performance comparison as discussed in the two previous sub-sections, the consistent performance comparison over five-fold CV and independent validation test demonstrates that the proposed Meta-iAVP could accurately discriminate AVPs from Non-AVPs on unknown peptides. In particular, its high MCC value indicates that this new AVP model could effectively reduce the number of both false positive (FP) and false negative (FN) as well as narrow down experimental efforts. As our proposed model outperformed the other existing methods, it is reasonable due to the following aspects: (i) amongst various types of features employed in this study, PseAAC and Am-PseAAC features are firstly employed in AVP prediction. Many studies reported that these two feature have been successfully implemented to predict many peptides and proteins [15,47,50,73,74,75,76,77,78,79]; (ii) the parameters of our proposed model were optimized by using the five-repeated five-fold CV indicating that our estimated parameters were more stable and accurate [80]; (iii) most of the existing predictors [41,42,44] were developed by using a combination of various types of features causing two outcomes: Information redundancy and the overfitting problem. On the other hand, we used only six-dimensional (6D) feature vectors that provided not only sufficient but also comprehensive information for AVP prediction; and (iv) our final meta-predictor was constructed by taking advantage of feature learning scheme. As seen in Table 4 and Table 5, the performance comparisons revealed that our proposed model is more effective and promising for AVP prediction.

### 2.6. Meta-iAVP web server

In an effort to maximize the utility of the prediction model by the scientific community, we have deployed the predictive model as a web server that is also called the Meta-iAVP (i.e., using the best model as described in previous sections). The web interface of the web server was established using the Shiny package under the R programming environment. The web server is freely accessible at http://codes.bio/meta-iavp/. Screenshots of the Meta-iAVP web server are shown in Figure 7 in which panel A shows the web server prior to submission of input data and panel B shows the web server after the prediction has been made.

Briefly, a step-by-step guide on using the web server is given below:**Step 1.** Proceed to entering the following URL into the web browser, http://codes.bio/meta-iavp/.**Step 2.** Users have the option of either entering the query peptide sequence directly into the Input box or uploading the sequence file by clicking on the “Choose file” button (i.e., found below the “Enter your input sequence(s) in FASTA format heading”).**Step 3.** Click on the *“*Submit” button in order to start the prediction process.**Step 4.** Once predictions are made, the results output are shown in the grey box found below the “Status/Output” heading. The prediction process requires only a few seconds to process. After predictions are made, the prediction output can be conveniently downloaded as a CSV file by pressing on the “Download CSV button”.

## 3. Materials and Methods

In practice, the prediction of peptide function is quite difficult and hard, particularly in dealing with a complicated biological system. Nevertheless, the development of an accurate prediction method might be deemed rewarding and successful if it could help provide some useful information. Thus, the present study was devoted to develop a new meta-predictor for discriminating AVPs from Non-AVPs in peptide sequences. To establish a really useful computational method for a biological system, we followed Chou’s five-step guidelines mentioned in [81,82,83,84,85]: (i) construct or collect a reliable dataset that is experimentally validated sequences for training and validating the model; (ii) represent peptides sequences that can truly reflect their intrinsic properties to be predicted; (iii) develop a powerful algorithm or engine to operate the prediction; (iv) evaluate the prediction method with appropriate and rigorous cross-validation tests; and (v) develop a user-friendly web-server for users that can easily get their desired result without needing to go through the mathematical and statistical details. Below, we describe in detail how to deal with these steps one by one. Furthermore, Figure 2 shows the workflow of Meta-iAVP which works in discriminating peptides as AVPs or Non-AVPs.

### 3.1. Dataset Preparation

One of the most important steps is to establish a reliable and stringent benchmark dataset to train and test the proposed method. To objectively evaluate the performance of the proposed method and fairly compare it with the existing methods [41,42,44], the same datasets, i.e., T^544p+407n^, T^544p+544n^, T^60p+45n^, and T^60p+60n^, which were obtained from the study by Thakur et al. [41] were taken as the benchmark dataset in this study. For training the prediction model, the two benchmark datasets T^544p+407n^ and T^544p+544n^ that were used in this study can be summarized by the following formula:(1)$$T^{544p+407n}=T^{544p}\cup T^{407n}$$
(2)$$\;T^{544p+544n}=T^{544p}\cup T^{407n}\;$$
where $T^{544p}$ and $T^{407n}$ represent collections of 544 and 407 experimentally validated AVP and Non-AVPs, respectively, while $T^{544n}$ represent a collection of 544 non-experimentally validated Non-AVPs and the symbol ∪ represents the union from the set theory. Meanwhile, for assessing the efficient ability in predicting unknown peptides, the independent validation datasets V^60p+45n^ and V^60p+60n^ were used to evaluate the prediction performance from the prediction model constructed by the datasets T^544p+407n^ and T^544p+544n^, respectively, summarized by the following formula:(3)$$V^{60p+45n}=V^{60p}\cup V^{45n}$$
(4)$$V^{60p+60n}=V^{60p}\cup V^{60\mathrm{nn}}$$
where $V^{60p}$ and $V^{45n}$ represent collections of 60 and 45 experimentally validated AVP and Non-AVPs, respectively, while $V^{60n}$ represent a collection of 60 non-experimentally validated Non-AVPs.

### 3.2. Feature Extraction of Peptides

In development of a sequence-based predictor for predicting the biological activity, the feature extraction process is one of the most crucial aspects where peptide sequences are represented in a way that can afford a comprehensive and proper descriptor of the features reflecting their biological activities. Given a peptide sequence (P), it can be represented as:(5)$$P=p_{1}p_{2}p_{3}\ldots p_{1N}$$
where $p_{i}$ and N denote the *i*th residue in the peptide P and the peptide length, respectively. To develop the sequence-based predictor based on machine learning models, five different compositions and properties (i.e., AAC, DPC, PseAAC, Am-PseAAC, and GDC) that cover various aspects of sequence information were used. These five features have been successfully used to predict many peptides and proteins, such as human leukocyte antigen gene [86,87]; protein crystallization [50,88], the oligomeric states of fluorescent proteins [89], the bioactivity of host defense peptides [48], human leukocyte antigen gene [86,87], antifreeze proteins [49], hemolytic activity of peptides [46], antihypertensive activity of peptides [47], and anti-angiogenic activity of peptides [74].

AAC and DPC are the proportions of each amino acid and dipeptide in a peptide sequence P that are expressed as fixed lengths of 20 and 400, respectively. Thus, in terms of AAC and DPC features, a peptide **P** can be expressed by vectors with 20D and 400D (dimension) spaces, respectively, as formulated by:(6)$$P={\left[{\mathrm{aa}}_{1},{\mathrm{aa}}_{2},\ldots ,\;{\mathrm{aa}}_{20}\right]}^{T}$$
(7)$$P={\left[{\mathrm{dp}}_{1},{\mathrm{dp}}_{2},\ldots ,\;{\mathrm{dp}}_{400}\right]}^{T}$$
where T is the transposed operator, while aa_1_, aa_2_…, aa_20_ and dp_1_, dp_2_…, dp_400_ are occurrence frequencies of the 20 and 400 native amino acids and dipeptides, respectively, in a peptide sequence P. As described, DPC is defined as the fraction of any two adjacent amino acids as a dipeptide pair. It could be stated that the information of non-adjacent amino acids might be lost. Thus, the GDC feature is developed to remedy such problem. This feature represents the number of occurrences of two amino acids that are separated by *g* gaps (i.e., *g* = 0 represents a DPC feature). In this work, *g* = 1, 2, 3, 4, and 5 was used.

As mentioned in previous studies [81,82,83] and shown in Equations (3)–(4), AAC, DPC and *g*-gap features only provide compositional information of a peptide sequence, but all the sequence-order information may be completely lost. To remedy this limitation, PseAAC and Am-PseAAC approaches were proposed by Chou [80,81]. According to Chou’s PseAAC, the general form of PseAAC for a peptide P is formulated by:(8)$$P={\left[\Psi_{1},\Psi_{2},\ldots ,\Psi_{u},\ldots ,\;\Psi_{\Omega }\right]}^{T}$$
where the subscript Ω is an integer to reflect the feature’s dimension. The value of Ω and the component of $\Psi_{u}$, where u=1,2,…,Ω is dependent on the protein or peptide sequences. In this study, the parameters of PseAAC (i.e., the discrete correlation factor λ and weight of the sequence information ϖ) were estimated by using the optimization procedure as described hereafter. The dimension of PseAAC feature is 20 + λ×ϖ. Since the hydrophobic and hydrophilic properties of proteins play an important role in the folding and interaction of proteins, Am-PseAAC was introduced by Chou [81]. The dimension of Am-PseAAC feature is 20 + 2λ. The first 20 components are the 20 basic AAC ($p_{1},p_{2},\ldots ,\;p_{20}$) while the next 2λ ones denote the set of correlation factors that reveal the physicochemical properties such as hydrophobicity and hydrophilicity (as) along a protein or peptide sequence as formulated by:(9)$$P={\left[p_{1},p_{2},\ldots ,\;p_{20},p_{20+\lambda },p_{20+\lambda +1},\ldots p_{20+2\lambda }\right]}^{T}$$

The concrete values of hydrophobicity and hydrophilicity are given in Table A1. In this study, the five aforementioned features of peptide sequences were generated by using the protr package in the R programming environment [90]. The parameters of PseAAC (weight^1^ and lamda^1^) and Am-PseAAC (weight^2^ and lamda^2^) were optimized by varying weight and lambda values from 0 to 1 and 1 to 10 with step sizes of 0.1 and 1, respectively, on the whole T^544p+407n^ and T^544p+544n^ datasets as assessed by a 5-fold CV procedure. More details of how to estimate such parameters can be found elsewhere [15,73,74,75].

### 3.3. Machine Learning Algorithms

The capability of prediction for the proposed model developed herein is dependent not only on the feature representation process but also on the selection of machine learning algorithms. This study exploited six popular and convenient ML algorithms, namely k-NN, rpart, glm, RF, XGB, and SVM, for discriminating AVPs from Non-AVPs. Previously, these ML algorithms have been extensively utilized in various domains [84,85,91,92,93,94,95,96,97,98,99]. In this study, the six ML algorithms were implemented using the caret package in the R software [100]. Herein, the b concept and associated parameter optimization for the six ML algorithm are given as follows:

The *k*-NN method is conceptually based on a distance function to measure the similarity between a pair of samples. This method is categorized as an instance-based learning algorithm that has been shown to be very effective for a variety of problem domains [86]. Given a dataset consisting of labeled peptide D, a positive integer k and an unknown peptide $P_{new}$, the *k*-NN classifier finds the *k* nearest neighbors of $P_{new}$ in the dataset D, called knn($P_{new}$), and returns the dominating class, i.e., AVPs or Non-AVPs, in knn(x) as the prediction result of label of the peptide $P_{new}$. Optimization of *k*-NN parameter (k) was determined by using the search space to maximize a five-fold CV accuracy on the benchmark datasets T^544p+407n^ and T^544p+544n^ are [5,23] with the step of two.

The rpart method has been developed since the 1980s [101]. This method uses recursive partitioning for classification, regression and survival trees. This method can be used to build classification or regression models using two main steps. Firstly, the single feature which provides the best split for the dataset into two groups is identified. After that, each dataset in further divided into two groups as a sub-group, and so on recursively until a particular stopping criterion is reached, i.e., either reaching a minimum size or on improvement can be made. The second step is to resample a dataset and trim back to full tree.

The glm method is one of the most useful ML algorithms used for classification and regression tasks, because it can be applied to many different types of domains. This method is a flexible generalization of ordinary linear regression that allows the output variables having error distribution models rather than a normal distribution. The glm method attempts to determine the relationship between a set of features and classes by fitting a linear equation to a dataset consisting of labeled peptide D. In the glm analysis, stepwise regression is used to select the most informative feature for improving the prediction performance. For rpart and glm methods, the default caret parameter setting was used [90].

RF was constructed according to the described original RF algorithm [101,102]. This model is an ensemble model consisting of many classification and regression tree (CART) classifiers to perform classification and regression tasks and improves prediction performances of CART classifiers by growing a number of weak CART classifiers. RF utilizes the concepts of bagging and random feature selection. The prediction result of the classification task is obtained by using a simple voting among outputs of all trees to get one final prediction. In regression, a final prediction is the average of prediction results of many decision trees. Herein, the RF classifier was established using the randomForest package in the R software [101]. To enhance the performance of the RF model, two parameters namely ntree (i.e., the number of tree used for constructing the RF classifier) and mtry (i.e., the number of random candidate features) were determined using the caret R package [100] with a five-fold CV approach. The search space of ntree and mtry are (100,500) and (1,10) with the steps of 100 and 1, respectively.

XGBoost is a meta-algorithm used to construct an ensemble of strong learners from weak learners, typically decision trees, on a modified dataset [103]. XGBoost, proposed by Chen and Guestrin [104] is a boosted tree algorithm, which follows the principle of gradient boosting. In recent years, XGBoost has been used extensively by data scientists and achieves satisfactory results on various biological problems [105]. In this study, the prediction of AVPs can be considered as a binary classification problem. Given a peptide sequence, we used XGBoost to predict its class label (−1 or 1), where +1 and −1 represent AVPs and Non-AVPs, respectively. For achieving the best XGBoost model, five parameters namely eta (i.e., the number of the learning rate), max_depth (i.e., the number of the depth of the tree), colsample_bytree (i.e., the number of features or variables to construct a learner), subsample (i.e., the number of samples or observations to construct a learner), and nrounds (i.e., the maximum number of iterations) were determined using the caret R package [100] with a five-fold CV approach. The search space of eta, max_depth, colsample_bytree, subsample and nrounds are (0.3, 0.4), (1,5), (0.6,0.8), (0.500, 1.000), and (50,250) with the steps of 0.1, 1, 0.2, 0.125, and 50, respectively.

SVM method is a well-known ML algorithm based on the Vapnik-Chervonenkis theory of statistical learning [106,107,108], which has been widely used in various biological problems [67,68,69,70,71,73,75,82,87,109,110]. The principle idea of this method is to map the original feature vectors having m-dimensional vector into a higher Hilbert space with n-dimensional vector, where m < n, and then determine a separating hyper plane with the largest distance between two classes. In this work, each sample on the benchmark datasets T^544p+407n^ and T^544p+544n^ has a corresponding label (−1 and 1) where +1 and −1 represent AVPs and Non-AVPs, respectively. Many studies reported that SVM can perform well on small sample size due to its excellent learning and best generalization abilities [73,75]. In this study, the *kernlab* R package [111] was used to implement the SVM model. To obtain an optimal SVM model, the regularization parameter *C* and kernel parameter γ were tuned by using grid search method with a cross-validation technique, of which the search space for *C* and γ are (2^–8^,2^8^) and (2^−8^, 2^8^) with steps of two and two, respectively.

### 3.4. Feature Importance Analysis

In this work, we performed the analysis and identification of feature importance for each type of sequence feature by using the RF method to provide a better understanding of the biophysical and biochemical properties of AVPs. In practice, the RF method provides two measures for ranking feature importance, i.e., the mean decrease of Gini index and the mean decrease of prediction accuracy. Since Calle and Urrea [112] demonstrated that the MDGI provided a more robust result as compared to the mean decrease of prediction accuracy, we utilized the MDGI value to rank the importance of interpretable features including AAC and DPC. The Gini index can be defined as MDGI is an impurity measure that corresponds to the ability of each feature in discriminating the sample classes. The Gini index can be defined as
(10)$$1-\sum_{c=1}^{2}p^{2}(c|t)$$
where $\sum_{c=1}^{2}p^{2}(c|t)$ denotes the estimated class probability for node *t* in a tree classifier and *c* is the class label (i.e., either AVP or Non-AVP). Features with the largest MDGI value is considered to be an important feature as it significantly contributes to the prediction performance. Herein, the MDGI values of feature importance for each type of sequence feature is estimated using the randomForest package in the R software [101].

### 3.5. Performance Evaluation

For the prediction problem, it is essential to determine the success and error rates of a given classifier. In practice, there are three CV methods which are traditional approaches, i.e., sub-sampling test or k-fold cross-validation (k-fold CV), jackknife test, and independent validation test or external test. Among these, the jackknife test is recognized as the least arbitrary and most objective one, as mention by equation 28–32 in Chou [81]. Meanwhile, the external test is considered as one of the most rigorous and objective methods for cross-validation in statistics. In k-fold cross-validation procedure, the training set is randomly separated into k subsets. From the k subsets, a single subset is taken as the testing set to validate the prediction model trained and learned by the remaining k-1 subsets. This process is repeated k times, until each subset had been used as the testing set. During the jackknifing process, a single sample in the whole dataset having N samples is taken as the testing set and the remaining N-1 samples are used for training the model. This process is repeated N times, until each sample has been used as the testing set.

In order to evaluate the prediction ability of the model, the following sets of four metrics are used as follows:(11)$$\mathrm{Ac}=\frac{\mathrm{TP}+\mathrm{TN}}{\left(\mathrm{TP}+\mathrm{TN}+\mathrm{FP}+\mathrm{FN}\right)}$$
(12)$$\mathrm{Sn}=\frac{\mathrm{TP}}{\left(\mathrm{TP}+\mathrm{FN}\right)}$$
(13)$$\mathrm{Sp}=\frac{\mathrm{TN}}{\left(\mathrm{TN}+\mathrm{FP}\right)}$$
(14)$$\mathrm{MCC}=\frac{\mathrm{TP}\times \mathrm{TN}-\mathrm{FP}\times \mathrm{FN}}{\sqrt{\left(\mathrm{TP}+\mathrm{FP}\right)\left(\mathrm{TP}+\mathrm{FN}\right)\left(\mathrm{TN}+\mathrm{FP}\right)\left(\mathrm{TN}+\mathrm{FN}\right)}}$$
where Ac, Sn, Sp, and MCC are called accuracy, sensitivity, specificity and Matthews coefficient correlation, respectively. TP, TN, FP, and FN represent the instances of true positive, true negative, false positive and false negative, respectively. In 2009, Kim [80] demonstrated that the repeated k-fold CV procedure yielded better performances than the non-repeated k-fold CV by reducing the variability of the model. In this study, the five-repeated five-fold CV in conjunction with an independent validation test are used to measure the performance of the model.

### 3.6. Feature Representation Learning

Previously, feature learning scheme has been successfully implemented to predict many peptides and proteins [68,69,70]. Therefore, in this study, the same protocol was utilized to generate a new feature representation, as illustrated in Figure 2. The procedures of this scheme are briefly described as follows:

#### 3.6.1. Constructing Initial Features

As mentioned above, each peptide sequence was extracted as a numerical representation based on AAC, PseAAC, Am-PseAAC, DPC, and GDC called initial features. The parameters of PseAAC (weight^1^ and lamda^1^) and Am-PseAAC (weight^2^ and lamda^2^) were optimized by varying weight and lambda values from 0 to 1 and 1 to 10 with step sizes of 0.1 and 1, respectively, on the benchmark datasets T^544p+407n^ and T^544p+544n^ as assessed by a five-fold CV procedure. In this study, values of weight^1^, weight^2^, lamda^1^, and lamda^2^ as performed on the benchmark datasets T^544p+407n^ and T^544p+544n^ are (0.6, 0.1, 3, and 4) and (0.6, 0.2, 4, and 3), respectively. Meanwhile, the parameter of GDC feature (*g*-gap) were optimized by choosing from one to five as assessed by a five-fold CV procedure. The optimum values of *g* on the benchmark datasets T^544p+407n^ and T^544p+544n^ are one and three, respectively.

#### 3.6.2. Constructing a New Feature Representation

Firstly, the initial features for each type of feature were exploited to train six ML models (i.e., *k*-NN, rpart, glm, RF, XGBoost, and SVM) using the two benchmark datasets and five-fold CV for generating the predicted label. Secondly, for each type of feature, the new feature representation O(M) was obtained by concatenating all the predicted labels from the six ML models. In our experiment, the predicted label is represented with either the value of 0 or 1, where 1 and 0 represent the predicted results as AVPs and Non-AVPs, respectively. Finally, for a given peptide sequence P, the sequence P is represented with a new 6D feature vector.

#### 3.6.3. Learning a New Feature for Meta-Predictor Representation

The new feature representations were used as input to train the RF model and subsequently used for formulating the final meta-predictor separately for the two benchmark datasets by means of the five-repeated five-fold CV.

### 3.7. Development of the Meta-iAVP Web Server

The best predictive model was deployed as a web server by harnessing the Shiny R package to craft the web interface. Firstly, the web server accepts as input the input sequence in FASTA format (i.e., either by from the input text box or from the uploaded FASTA file). Secondly, upon submission of the input sequence by invoking the Submit button, the query sequences are subjected to descriptor calculation and subsequently applied to the predictive model described previously. The resulting prediction of the class labels (i.e., as either AVP or Non-AVP) along with their probability values are displayed in the prediction output box. Results from the prediction process is also provided as a CSV file upon invoking the Download button found directly underneath the output box.

## 4. Conclusions

Owing to the medical significance and potential utility of AVPs as promising antiviral drug candidates, there is intensive efforts in the development of computational models for rapidly and accurately identifying AVPs on unknown peptides. In this study, we have developed a novel meta-predictor for AVP prediction called the Meta-iAVP. In constructing this meta-predictor, a feature representation learning scheme based on six different ML algorithms and five feature types were applied in model construction. Experimental results demonstrated the superiority of the proposed Meta-iAVP model based on the feature representation learning scheme over models constructed by the aforementioned ML algorithms and features. Furthermore, to confirm the effectiveness of the Meta-iAVP model, we have also performed comparative analyses with other state-of-the-art AVP predictors. It was observed from rigorous five-fold cross-validation and independent validation test that the proposed model was more effective and promising for AVPs prediction. To maximize the convenience of the vast majority of experimental scientists, the model was deployed as a web server that also goes by the same name, Meta-iAVP, which has been made freely available at http://codes.bio/meta-iavp/. It is anticipated that Meta-iAVP will serve as a useful, high throughput and cost-effective tool for large-scale analysis of AVPs that would help contribute to a series of interesting follow-up research studies involving antiviral peptides and other related therapeutic peptides. Although, Meta-iAVP displayed a superior performance over that of existing methods as assessed by rigorous cross-validation methods, there is still room for further improvements. For example, to improve the usefulness and efficacy for drug development and experimental research, we will make an effort to develop a computational model for predicting the inhibition of specific viruses in future studies.

## Figures and Tables

**Figure 1 ijms-20-05743-f001:**
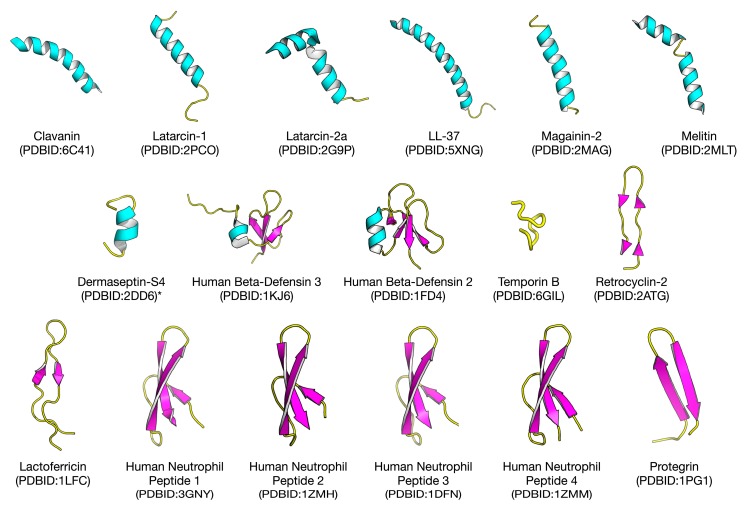
Structures of selected antiviral peptides that have been experimentally elucidated. Each structure is labelled by a common name followed by the Protein Data Bank Identification number (PDBID) in parenthesis on the subsequent line. * Dermaseptin-S4: The structure and available PDBID is that of a truncated peptide, which was experimentally tested to be effective.

**Figure 2 ijms-20-05743-f002:**
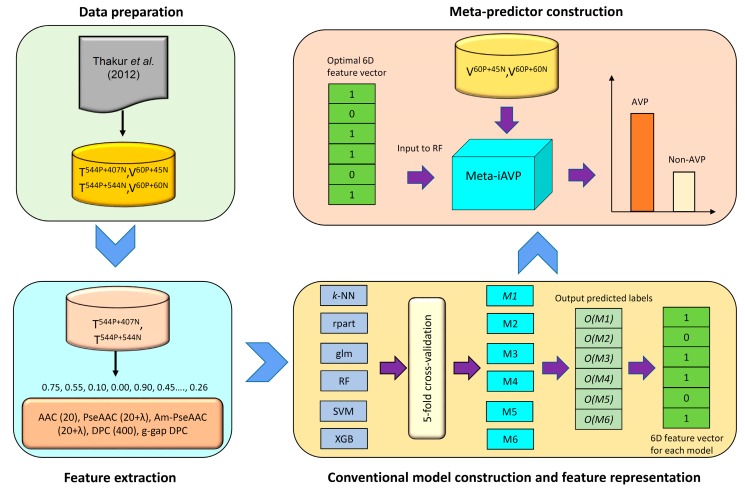
Schematic framework of Meta-iAVP. Overview of the proposed methodology for discriminating AVPs from Non-AVPs involving the following steps: (1) preparing two benchmark datasets; (2) extracting a peptide sequence with five types of features to encode six models; (3) constructing six ML models to generate a six-dimentional feature for each type of feature O(M), where 1 and 0 are represented with AVPs and Non-AVPs, respectively; and (4) establishing the meta-predictor for each benchmark dataset that separates a query peptide into AVPs and Non-AVPs.

**Figure 3 ijms-20-05743-f003:**
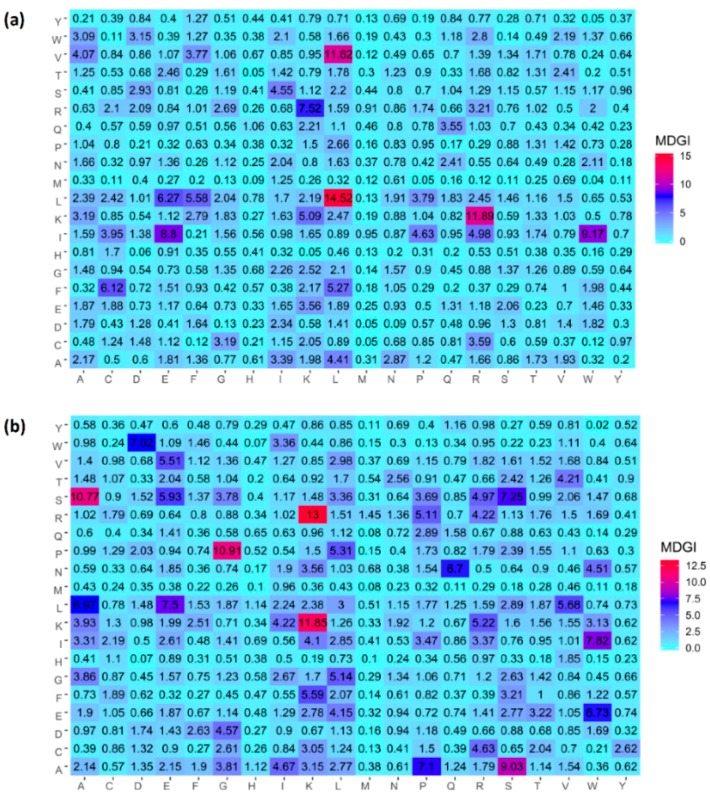
Heat map of the mean decrease of Gini index of dipeptide compositions for the T^544p+407n^ + V^60p+45n^ (**a**) and T^544p+544n^ + V^60p+60n^ (**b**) datasets. It should be noted that features with the largest value of MDGI are deemed to be the most important.

**Figure 4 ijms-20-05743-f004:**
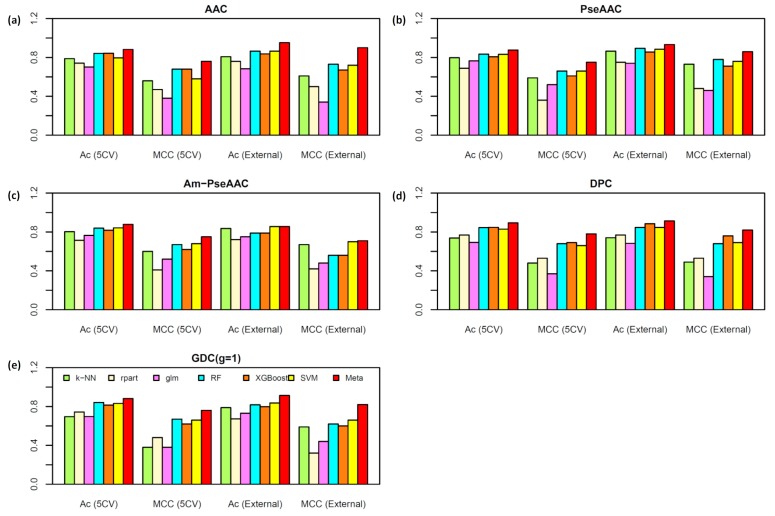
Performance comparisons of AVP predictors based on different six machine learning algorithms types of features, i.e., AAC (**a**), PseAAC (**b**), Am-PseAAC (**c**), DPC (**d**), and GDC (**e**), on the dataset T^544p+407n^ + V^60p+45n^, respectively.

**Figure 5 ijms-20-05743-f005:**
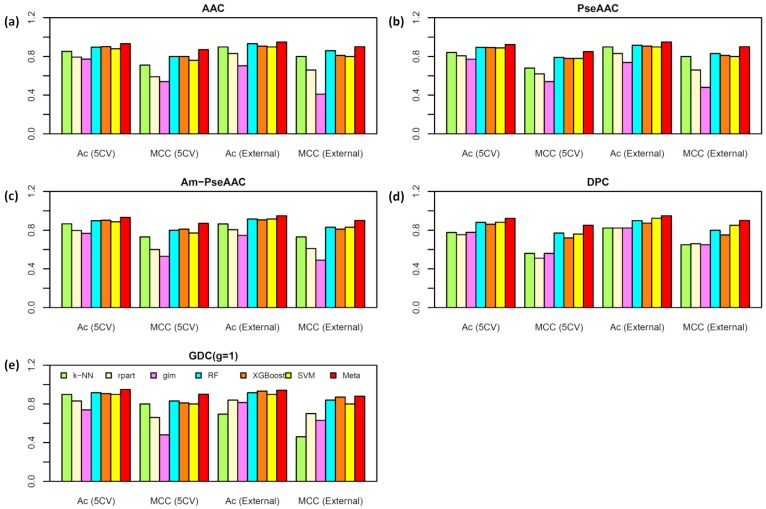
Performance comparisons of AVP predictors based on different six machine learning algorithms types of features, i.e., AAC (**a**), PseAAC (**b**), Am-PseAAC (**c**), DPC (**d**), and GDC (**e**), on the dataset T^544p+544n^ + V^60p+60n^, respectively.

**Figure 6 ijms-20-05743-f006:**
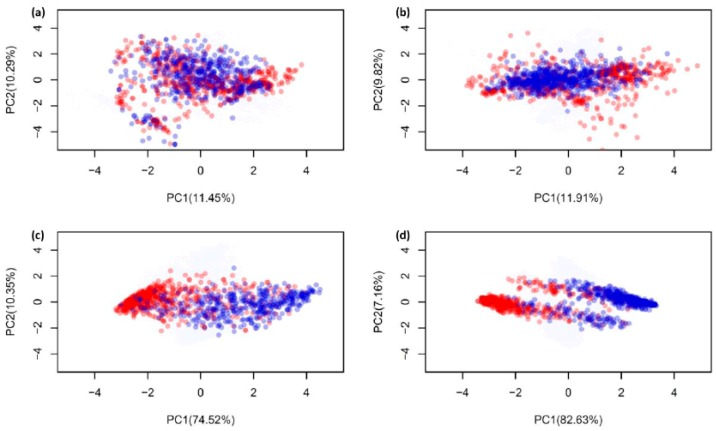
Principle component analysis (PCA) scores plot of the distribution of AVPs and Non-AVPs, where AVPs and Non-AVPs are represented by red and blue circles, respectively. (**a**) and (**c**) represent the distribution of amino acid composition and a new feature representation, respectively, obtained from the dataset T^544p+407n^ + V^60p+45n^, while (**b**) and (**d**) represent the distribution of pseudo amino acid composition and a new feature representation, respectively, obtained from the dataset T^544p+544n^ + V^60p+60n^.

**Figure 7 ijms-20-05743-f007:**
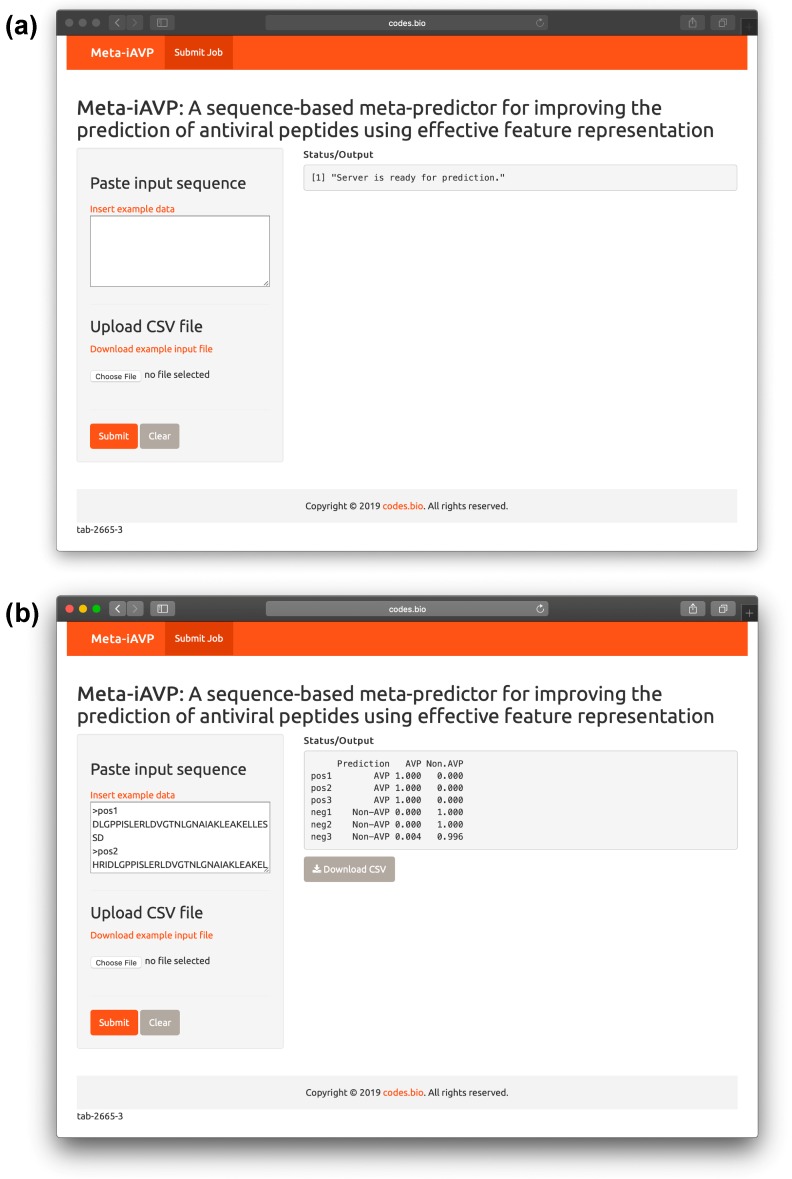
Screenshots of the Meta-iAVP web server before (**a**) and after (**b**) submission of sequence data for prediction. Predictions are shown alongside the probability values for each class predictions. Results are provided as a downloadable CSV file by clicking on the gray button underneath the prediction output.

**Table 1 ijms-20-05743-t001:** Summary of existing methods for predicting antiviral peptides.

Method	Classifier ^a^	Sequence Feature ^b^	Stand-Alone Program	Webserver
AVPpred [41]	SVM	AAindex	−	✓
Chang et al.’s method [42]	RF	AAC, aggregation	−	−
AntiVPP 1.0 [44]	RF	PCP	✓	−
Meta-iAVP (This study)	Meta-predictor	AAC, Am-PseAAC	−	✓

^a^ RF: Random forest and SVM: Support vector machine. ^b^ AAC: Amino acid composition, AAindex: Amino Acid index database, aggregation: Aggregation propensity, Am-PseAAC: Amphiphilic pseudo amino acid composition, and PCP: Physicochemical properties.

**Table 2 ijms-20-05743-t002:** Amino acid compositions (%) of AVP and Non-AVP along with their mean decrease of Gini index (MDGI) values on T^544p+407n^ dataset.

Amino Acid	AVP (%)	Non-AVP (%)	Difference	*p*-Value	MDGI
**K-Lys**	0.092	0.078	0.014	<0.05	49.27(1)
**T-Thr**	0.032	0.055	−0.023	<0.05	46.27(2)
**L-Leu**	0.119	0.09	0.029	<0.05	35.06(3)
**I-Ile**	0.068	0.046	0.022	<0.05	34.52(4)
**S-Ser**	0.054	0.057	−0.003	0.464	30.95(5)
**W-Trp**	0.049	0.024	0.025	<0.05	30.93(6)
**N-Asn**	0.04	0.049	−0.009	<0.05	30.19(7)
**R-Arg**	0.079	0.082	−0.003	0.685	28.52(8)
**C-Cys**	0.038	0.035	0.003	0.499	26.33(9)
**E-Glu**	0.062	0.051	0.011	<0.05	24.87(10)
**D-Asp**	0.038	0.042	−0.004	0.204	22.93(11)
**A-Ala**	0.074	0.079	−0.005	0.384	21.85(12)
**V-Val**	0.049	0.062	−0.013	<0.05	21.1(13)
**P-Pro**	0.033	0.054	−0.021	<0.05	19.73(14)
**Q-Gln**	0.036	0.036	0	0.916	17.84(15)
**G-Gly**	0.047	0.059	−0.012	<0.05	17.25(16)
**H-His**	0.016	0.022	−0.006	<0.05	14.9(17)
**F-Phe**	0.041	0.038	0.003	0.358	14.49(18)
**Y-Tyr**	0.021	0.03	−0.009	<0.05	12.09(19)
**M-Met**	0.011	0.014	−0.003	0.085	6.27(20)

**Table 3 ijms-20-05743-t003:** Amino acid compositions (%) of AVP and Non-AVP along with their MDGI values on T^544p+544n^ dataset.

Amino Acid	AVP (%)	Non-AVP (%)	Difference	*p*-Value	MDGI
**K-Lys**	0.092	0.046	0.045	<0.05	77.11(1)
**P-Pro**	0.033	0.068	−0.035	<0.05	68.87(2)
**C-Cys**	0.038	0.022	0.015	<0.05	57.68(3)
**T-Thr**	0.032	0.053	−0.021	<0.05	46.84(4)
**S-Ser**	0.054	0.083	−0.029	<0.05	39.57(5)
**W-Trp**	0.049	0.015	0.033	<0.05	36.83(6)
**V-Val**	0.049	0.069	−0.02	<0.05	25.69(7)
**A-Ala**	0.074	0.087	−0.013	<0.05	24.40(8)
**G-Gly**	0.047	0.072	−0.025	<0.05	24.25(9)
**L-Leu**	0.119	0.117	0.002	0.728	23.80(10)
**I-Ile**	0.068	0.042	0.026	<0.05	23.42(11)
**H-His**	0.016	0.021	−0.005	<0.05	23.13(12)
**E-Glu**	0.062	0.056	0.006	0.108	20.13(13)
**Q-Gln**	0.036	0.04	−0.004	0.18	18.50(14)
**N-Asn**	0.04	0.03	0.01	<0.05	18.48(15)
**R-Arg**	0.079	0.061	0.018	<0.05	17.67(16)
**F-Phe**	0.041	0.038	0.003	0.321	16.57(17)
**D-Asp**	0.038	0.038	0	0.982	15.75(18)
**Y-Tyr**	0.021	0.023	−0.001	0.537	10.57(19)
**M-Met**	0.011	0.017	−0.006	<0.05	10.33(20)

**Table 4 ijms-20-05743-t004:** Performance comparisons between Meta-iAVP and the six machine learning algorithms as assessed by the five-repeated five-fold cross-validation and independent validation tests.

Dataset	Method ^a^	Ac (%)	Sn (%)	Sp (%)	MCC
T^544p+407n^	*k*-NN	78.79	88.24	66.13	0.56
	rpart	74.09	81.03	64.82	0.47
	glm	70.15	82.87	53.27	0.38
	RF	84.22	85.70	82.34	0.68
	XGBoost	84.33	86.69	80.97	0.68
	SVM	79.53	83.81	73.86	0.58
	Meta-predictor	88.17	89.23	86.94	0.76
T^544p+544n^	*k*-NN	84.15	82.53	86.07	0.68
	rpart	80.63	82.37	79.73	0.62
	glm	77.11	77.78	76.78	0.54
	RF	89.44	84.18	94.68	0.79
	XGBoost	89.16	87.48	90.90	0.78
	SVM	88.79	87.13	90.71	0.78
	Meta-predictor	92.31	88.44	96.16	0.85
V^60p+45n^	*k*-NN	80.77	95.00	61.36	0.61
	rpart	75.96	86.67	61.36	0.50
	glm	68.27	86.67	43.18	0.34
	RF	86.54	86.67	86.36	0.73
	XGBoost	83.65	85.00	81.82	0.67
	SVM	86.54	93.33	77.27	0.72
	Meta-predictor	95.19	96.67	93.18	0.90
V^60p+60n^	*k*-NN	89.83	85.00	94.83	0.80
	rpart	83.05	88.33	77.59	0.66
	glm	73.73	78.33	68.97	0.48
	RF	91.53	90.00	93.10	0.83
	XGBoost	90.68	90.00	91.38	0.81
	SVM	89.83	88.33	91.38	0.80
	Meta-predictor	94.92	93.33	96.55	0.90

^a^*k*-NN: *k*-nearest neighbor, rpart: ecursive partitioning and regression trees, glm: Generalized linear model, RF: Random forest, XGBoost: Extreme gradient boosting, and SVM: Support vector machine.

**Table 5 ijms-20-05743-t005:** Performance comparisons between Meta-iAVP and the three existing methods as assessed by the five-repeated five-fold cross-validation and independent validation tests.

Dataset	Method ^a^	Ac (%)	Sn (%)	Sp (%)	MCC
T^544p+407n^	AVPpred	85.00	82.20	**88.20**	0.70
	Chang et al.’s method	85.10	86.60	83.00	0.70
	AntiVPP 1.0	-	-	-	-
	Meta-iAVP	**88.20**	**89.20**	86.90	**0.76**
T^544p+544n^	AVPpred	90.00	**89.70**	90.30	0.80
	Chang et al.’s method	91.50	89.00	94.10	0.83
	AntiVPP 1.0	-	-	-	-
	Meta-iAVP	**93.20**	89.00	**97.40**	**0.87**
V^60p+45n^	AVPpred	85.70	88.30	82.20	0.71
	Chang et al.’s method	89.50	91.70	86.70	0.79
	AntiVPP 1.0	-	-	-	-
	Meta-iAVP	**95.20**	**96.70**	**93.20**	**0.90**
V^60p+60n^	AVPpred	92.50	**93.30**	91.70	0.85
	Chang et al.’s method	93.30	91.70	95.00	0.87
	AntiVPP 1.0	93.00	87.00	97.00	0.87
	Meta-iAVP	**94.90**	91.70	**98.30**	**0.90**

^a^ Results were reported from the works of AVPpred, Chang et al.’s method, and AntiVPP 1.0. The highest values for each performance measure are shown in bold.

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
