# Peer review of "Meta-iAVP: A Sequence-Based Meta-Predictor for Improving the Prediction of Antiviral Peptides Using Effective Feature Representation"

_ijms, 2019, doi:10.3390/ijms20225743_

Round 1

Reviewer 1 Report

The manuscript  by  Shoombuatong etc. describes the development of a model, called Meta-iAVP, to predict the effectiveness of AVPs. This work provided a convenient tool for AVPs study. This model seems to have advantage in accuracy and effectiveness compared to existing predictors. I recommend for publication in IJMS and have the following suggestions.

The authors should check unintended grammar errors. I suggest to add a summary of what viruses peptide-based antivirals/ AVPs can inhibit and how effective they are in the introduction. Viruses are different. If this model can be developed to predict the inhibition of a specific virus, it will be of much more value.

Author Response

Point 1: The manuscript by Shoombuatong etc. describes the development of a model, called Meta-iAVP, to predict the effectiveness of AVPs. This work provided a convenient tool for AVPs study. This model seems to have advantage in accuracy and effectiveness compared to existing predictors. I recommend for publication in IJMS and have the following suggestions.

Response 1: Thank you for the encouraging and kind words of inspiration

Point 2: The authors should check unintended grammar errors. I suggest to add a summary of what viruses peptide-based antivirals/ AVPs can inhibit and how effective they are in the introduction. Viruses are different. If this model can be developed to predict the inhibition of a specific virus, it will be of much more value.

Response 2: Thank you for your suggestions. The unintended grammatical errors have been corrected within the manuscript. A brief mention of antiviral peptide-based drugs and their targets were already mentioned in the introduction at lines 83-88. In addition, more information has also been provided at lines 88-93. Furthermore, we agree with the recommendation of targeting this model towards a specific virus, which we are currently working on as part of future endeavours and is at the moment beyond the scope of this manuscript. For the development of prediction model for predicting the inhibition of a specific virus, it is the great idea for the drug development and basic research, but, in this study, we aimed to develop a model for discriminating AVPs from Non-AVPs. In the future study, we will make an effort to develop a computational model for predicting the inhibition of specific viruses in future studies (added in the Conclusion section).

Reviewer 2 Report

The authors of the manuscript" Meta-iAVP: A Sequence-Based Meta-Predictor for Improving the Prediction of Antiviral Peptides Using Effective Feature Representation" describe a method for the identification of anti-viral peptides. The method is interesting, and seems to outperform existing methods.
Most of the manuscript is well written, but the text should be carefully checked again, some sentences are awkwardly formulated and thus difficult to understand, in addition to some typos.

Numbers in text and tables should be rounded to one decimal after comma.

Some specific comments:

line 46 : I think >200 viruses is meant in stead of "<200 viruses"

Line 156: The listing of the 10 aa for both datasets is not very readable. Consider a table.

Line 264: From figures 4 and 5, I can not see why AAC and DPC are more important compared to the other three predictors. Please explain.

Table 4: Why is MCC not in % ?

Line 293: I do not understand what is meant by "obtained from our meta-predictor are 4-9% and 9-17% which are higher...etc."

Line 299: What is meant by "the method that takes new feature represention as the input feature"? What is the 'new feature'?
lime 300: 'predcition'?
Line 301: 'mentaioned' ?
line 302: 'effiectiveness' ?
line 326: How is the PCA performed? What do the numbers on the axis in fig 6 represent?

Fig 6: Why are the AAC + new feature only shown for dataset T544p+407n + V60p+45n shown and not for AAC + new feature on dataset T544p+544n + V60p+60n?
Similar, for PseAAC + new feature, where is the plot for dataset T544p+407n + V60p+45n?
Why is there a split in the distribution in 6d?

Line 367: FP and FN (fals positive and fals negative?) should be explained here.

Concerning the webserver: Additional information in needed. Why do I get 6 lines if only one sequence is uploaded? What does the outcomes pos1 and neg1 mean, etc.
A manual and link to a reference should be provided.

Line 478: Which values were used for the hydrophobicity and hydrophilicity? Maybe put these values (and other amino acid data used in an S.I.)

Author Response

Point 1: The authors of the manuscript" Meta-iAVP: A Sequence-Based Meta-Predictor for Improving the Prediction of Antiviral Peptides Using Effective Feature Representation" describe a method for the identification of anti-viral peptides. The method is interesting, and seems to outperform existing methods. Most of the manuscript is well written, but the text should be carefully checked again, some sentences are awkwardly formulated and thus difficult to understand, in addition to some typos.

Response 1: Thank you for the encouraging and kind words of inspiration. As suggested, the error has been corrected.

Point 2: Numbers in text and tables should be rounded to one decimal after comma.

Response 2: Thank you for the kind suggestion. By using 1 decimal point, all performance increase or decrease less than 10 % would not be detected by using 1 decimal point. However, we believe that at least 2 decimal point is required to observe model improvement especially when the performance increase or decrease is less than 10 %.

Point 3: line 46 : I think >200 viruses is meant in stead of "<200 viruses"

Response 3: Thank you for kind suggestion. As suggested, this error was unintentional and the correction has now been made.

Point 4: Line 156: The listing of the 10 aa for both datasets is not very readable. Consider a table.

Response 4: As suggested, the orders of AAs in Table 2 (Page 6) and 3 (Page 7) have been re-ranked according to the MDGI values.

Point 5: Line 264: From figures 4 and 5, I can not see why AAC and DPC are more important compared to the other three predictors. Please explain.

Response 5: Thank you for your kind comment of this point. ACC and DPC features did not afford better performance than other three predictors but they provide more interpretability for discriminating AVPs from Non-AVPs, which is helpful for biologists in designing novel peptides.

Point 6: Table 4: Why is MCC not in % ?

Response 6: Due to Eq. 14 on Page 19, MCC is usually represented in the range from −1 to 1.

Point 7: Line 293: I do not understand what is meant by "obtained from our meta-predictor are 4-9% and 9-17% which are higher...etc."

Response 7: Based on this point, the above mentioned sentence was changed as “obtained from the meta-predictor are 4-9% and 9-17%, respectively, which are higher than those resulting from k-NN, rpart, glm, RF, XGBoost and SVM models on” (on Page 10 at Line 291-293). This sentence means that the meta-predictor outperforms than k-NN, rpart, glm, RF, XGBoost and SVM models with the improvements of 4-9% and 9-17% on overall Ac and MCC values.

Point 8: Line 299: What is meant by "the method that takes new feature represention as the input feature"? What is the 'new feature'?

Response 8: As mentioned in the section Feature representation learning (Page 19), the new feature is the predicted labels from the six ML models (i.e. k-NN, rpart, glm, RF, XGBoost and SVM), where the predicted label is represented with either the value of 0 or 1 for AVPs and Non-AVPs, respectively.

Point 9: lime 300: 'predcition'?
Line 301: 'mentaioned' ?
line 302: 'effiectiveness' ?

Response 9: As suggested, the error has been corrected.

Point 10: line 326: How is the PCA performed? What do the numbers on the axis in fig 6 represent?

Response 10: Thank you for the great suggestions. More details of PCA analysis have been added on Page 12 at Line 333-334. Meanwhile, the numbers on the axis in Figure 6 have been described on Page 12 at Line 335-337

Point 11: Fig 6: Why are the AAC + new feature only shown for dataset T544p+407n + V60p+45n shown and not for AAC + new feature on dataset T544p+544n + V60p+60n?
Similar, for PseAAC + new feature, where is the plot for dataset T544p+407n + V60p+45n?
Why is there a split in the distribution in 6d?

Response 11: As mentioned in sections 2.2 and 2.3 (on Page 8-10), AAC and PseAAC features are the most important features for AVP prediction on T544p+407n + V60p+45n and T544p+544n + V60p+60n datasets, respectively. So, in the section 2.3, we aimed to compare between the discriminative ability of AAC versus the new feature and PseAAC versus new feature performed on T544p+407n + V60p+45n and T544p+544n + V60p+60n datasets, respectively. For the distribution of AVP and Non-AVP as shown in Fig 6d, overlapping points are less than that shown in Fig 6b, where almost all red and blue points are found in the left- and right-hand sides, respectively. Furthermore, the percentage of variance of the first two PCs of this figure was as high as 90%.

Point 12: Line 367: FP and FN (fals positive and fals negative?) should be explained here.

Response 12: Thank you for kind suggestion. FP and FN have been defined on Page 13 at Line 367.

Point 13: Concerning the webserver: Additional information in needed. Why do I get 6 lines if only one sequence is uploaded? What does the outcomes pos1 and neg1 mean, etc.
A manual and link to a reference should be provided.

Response 13: Thank you for your kind suggestion. We have fixed the webserver and now it returns the correct predictions. Before there was a bug where it returns the predictions from the example file. The peptide sequences labeled as pos1 and neg1 as used in the example input were obtain by random selection from the two benchmark datasets to represent peptides that are antiviral peptide (positive) and non-antiviral peptides (negative). These example input sequences are provided as initial examples in which the user can test the functionality of the web server where they can see that the positive (antiviral peptides) input sequence are indeed predicted to be positive (antiviral peptides).

Point 14: Line 478: Which values were used for the hydrophobicity and hydrophilicity? Maybe put these values (and other amino acid data used in an S.I.)

Response 14: As suggested, the values of the hydrophobicity and hydrophilicity for 20 amino acids are listed in Table A1 (on Page 20-21).